# Aberrant DNA Methylation of NPTX2 as an Indicator of Malignant Behavior in Thymic Epithelial Tumors [note 1]

**DOI:** 10.3390/cancers16020329

**Published:** 2024-01-11

**Authors:** Kazuya Kondo, Kyoka Muguruma, Shiho Soejima, Chikako Takai, Koichiro Kenzaki, Naoya Kawakita, Hiroaki Toba, Hiromitsu Takizawa

**Affiliations:** 1Department of Oncological Medical Services, Graduate School of Biomedical Sciences, Tokushima University, Tokushima 770-8509, Japan; qpdr8002@gmail.com (K.M.); mxat4254@yahoo.co.jp (S.S.); tomoamachie@yahoo.co.jp (C.T.); 2Department of Chest and Breast Surgical Oncology, Takamatsu Red Cross Hospital, Takamatsu 760-0017, Japan; kenzaki-koitiro@takamatsu.jrc.or.jp; 3Department of Thoracic, Endocrine Surgery and Oncology, Graduate School of Biomedical Sciences, Tokushima University, Tokushima 770-8503, Japan; kawakita.naoya@tokushima-u.ac.jp (N.K.); ht1109@tokushima-u.ac.jp (H.T.); takizawa@tokushima-u.ac.jp (H.T.)

**Keywords:** NPTX2, DNA methylation, mRNA expression, thymoma, thymic carcinoma, neuroendocrine tumors of the thymus

## Abstract

**Simple Summary:**

Metastatic or inoperable thymic carcinoma (TC) has a very poor prognosis. Since this tumor is rare, few clinical trials have been conducted to date, and thus, the development of new drugs is slow. The aberrant pathways involved in TC need to be examined in more detail in order to identify novel target molecules for TC therapy. Next-generation sequencing has been used to comprehensively investigate genetic alterations in thymic epithelial tumors (TET). However, limited information is currently available on epigenetic alterations. In the present study, genome-wide screening was conducted on aberrantly methylated CpG islands in TET, and 92 genes were identified. We then focused on the DNA methylation and mRNA and protein expression of *NPTX2* in TET. The DNA methylation level of NPTX2 was higher, while its mRNA and protein expression levels were lower in TC than in the normal thymus and thymoma. Furthermore, relapse-free survival was shorter in patients with high *NPTX2* DNA methylation levels than in those with low DNA methylation levels. Collectively, the present results indicate the potential use of NPTX2 as a tumor suppressor in TC.

**Abstract:**

Thymic epithelial tumors (TET) consist of thymomas, thymic carcinoma (TC), and neuroendocrine tumors of the thymus (NECTT). Genetic and epigenetic alterations in TET have been the focus of recent research. In the present study, genome-wide screening was performed on aberrantly methylated CpG islands in TET, and this identified neuronal pentraxin 2 (NTPX2) as a significantly hypermethylated CpG island in TC relative to thymomas. NPTX2 is released from pre-synaptic cells in response to neuronal activity/seizure, and plays a role in host immunity and acute inflammation. TET samples were obtained from 38 thymomas, 25 TC, and 6 NECTT. The DNA methylation, mRNA, and protein expression levels of *NPTX2* were examined. The DNA methylation rate of the *NPTX2* gene was significantly higher in TC than in the normal thymus and thymomas, except B3. The mRNA expression level of *NPTX2* was lower in TC than in the normal thymus. An inverse relationship was observed between mRNA expression levels and methylation levels. Relapse-free survival was shorter in patients with high *NPTX2* DNA methylation levels than in those with low DNA methylation levels. NECTT showed very high mRNA and protein expression levels and low DNA methylation levels of NPTX2. NPTX2 may function as a tumor suppressor in TC, and have an oncogenic function in NECTT.

## 1. Introduction

Thymic epithelial tumors (TET) are rare, and consist of thymoma, thymic carcinoma (TC), and neuroendocrine tumors of the thymus (NECTT) [1,2]. Thymomas are low-grade malignant tumors that are associated with myasthenia gravis and other autoimmune diseases [3]. They account for 75–80% of TET and 5-year survival rates are approximately 90% [4]. TC is an aggressive tumor and is not related to autoimmune diseases [3]. It accounts for 10–20% of TET and has a 5-year survival rate of approximately 55% [4].

The optimal treatment for TET depends on the clinical stage. Surgical resection remains the mainstay of treatment for patients with early- or moderate-stage TET (stages I, II, and III). Patients with advanced or relapsing tumors require chemotherapy and/or radiotherapy. However, metastatic and inoperable TC are associated with a poor prognosis [1,5,6,7].

Since the incidence of these tumors is very low, few clinical studies have been conducted, which has, in turn, impeded drug development. Targeted therapies and immunomodulatory agents for TET have been the focus of recent studies [8], and the findings obtained demonstrate the efficacy of sunitinib (an oral tyrosine kinase inhibitor of VEGFR, KIT, and PDGFR), lenvatinib (a new multi-targeted inhibitor of VEGFR, FGFR, RET, c-Kit, and other kinases), and pembrolizumab (an immune checkpoint inhibitor) for TC [9,10,11]. Further studies on the aberrant pathways in TC are warranted, and will contribute to the identification of novel molecules for targeted therapies.

A comprehensive examination of genetic alterations in TET using next-generation sequencing revealed mutations in TP53, TET2, CYLD, BRD7, SETD2, and CDKN2A in TC and GTF2I in thymomas [12,13,14,15]. Genomic alterations were also investigated in a large cohort of patients with TET, and their frequency was found to be significantly higher in TC than in thymomas, with mutations in CDKN2A (39.9%), TP53 (30.2%), and CDKN2B (24.6%) being more frequent in TC than in thymomas (TP53 (7.8%), DNMT3A (6.8%), and CDKN2A (5.8%)) [16].

However, epigenetic alterations in TET have not yet been examined in detail [15,16,17,18]. The findings of our previous studies demonstrate that the frequency of aberrant DNA methylation in four cancer-related genes (DAPK, p16, MGMT, and HPP1) was significantly higher in TC than in thymomas [17,18].

Furthermore, in our genome-wide screening of aberrantly methylated CpG islands (CGI) in TET, 92 genes were found to be significantly hypermethylated in TC [19]. We focused on five of these genes (GNG4, GHSR, HOXD9, SALL3, and GAD1) and showed significantly higher DNA methylation in TC than in thymomas [20,21].

Our genome-wide screening identified neuronal pentraxin 2 (NPTX2) as the 33rd significantly hypermethylated CGI in TC in relation to B3 thymomas [19].

Pentraxins are an evolutionarily conserved superfamily of proteins that are broadly grouped into the short and long pentraxin subfamilies [22]. The former comprises C-reactive protein, known as pentraxin 1, and the serum amyloid P component, called pentraxin 2 (PTX2) [23,24], while the latter includes neuronal pentraxins (NPTX1 and NPTX2) [22,25]. NPTX2 is a secreted glycoprotein that is widely expressed in the brain [26,27].

NPTX2 is released from pre-synaptic cells in response to neuronal activity/seizure. It may form a complex with NPTX1 at the post-synaptic membrane. The NPTX2 complex then aggregates a-amino-3-hydroxy-5-methylisoxazole-4-propionic acid (AMPA) receptors and neuronal pentraxin receptors. The internalization of the neuronal pentraxin complex and associated AMPA receptors occurs via endocytosis, and appears to be a protective mechanism against excitotoxicity [28]. A previous study reported the involvement of NPTX2 in host immunity and acute inflammation [29], while another showed that it played an important role in the development of central nervous system diseases, including Parkinson’s and Alzheimer’s diseases [30]. Recent studies suggested that the aberrant DNA methylation and mRNA and protein expression of NPTX2 affect the progression and prognosis of several cancers [31,32].

Therefore, we herein examined the DNA methylation and mRNA and protein expression levels of NPTX2, as well as their prognostic significance, in patients with TET.

## 2. Materials and Methods

### 2.1. Patients and Tissue Samples

TET tissues were collected during surgery or biopsy from 69 patients treated at Tokushima University Hospital (Tokushima, Japan) between 1985 and 2017. Tumor tissue and paired normal thymic tissue distant from the tumor were both snap-frozen and stored at −80 °C for later nucleic acid extraction.

DNA extracted from 64 tumors and 24 thymic tissues was subjected to a DNA methylation analysis. All 38 thymomas and 24 thymic tissue samples were obtained from frozen tissues. Due to the limited number of frozen samples of TC (n = 15) and NECTT (n = 3), formalin-fixed and paraffin-embedded (FFPE) samples of TC (n = 10) and NECTT (n = 3) were subjected to a DNA methylation array. RNA extracted from 48 tumor samples and 16 thymic tissue samples was used for RT-PCR. Immunohistochemistry (IHC) was performed on the FFPE materials of 39 samples (Table 1 and Table A1 (in Appendix A)).

All TET were classified according to the WHO histological classification system [2] and the criteria of the Masaoka–Koga staging system [33]. There were 38 cases of thymoma (6 of type A thymoma, 3 of type AB, 6 of type B1, 11 of type B2, and 12 of type B3), 25 of TC, and 6 of NECTT (4 of typical carcinoid and 2 of atypical carcinoid). Table 1 and Table A1 show the clinical and pathological characteristics of patients (age, sex, presence or absence of myasthenia gravis, the Masaoka–Koga stage, and WHO histological classification) whose samples were examined using pyrosequencing, RT-PCR, and IHC analyses.

The present study was conducted according to the tenets of the Declaration of Helsinki and approved by the local Ethics Committee (Tokushima University Hospital, approval number 2205-4). All patients provided their informed consent.

### 2.2. Global Methylation Analysis

In our previous study, we examined aberrantly methylated CGI in 7 TC- and 8 thymoma-type B3 samples (frozen specimens) using Illumina HumanMethylation450 K BeadChip, Santa Clara, CA, USA. In comparisons of TC with thymoma, 92 CGI were identified as significantly hypermethylated in TC samples (FDR < 0.05 and β-difference (TC—B3 thymoma) > 0.5), and the *NPTX2* gene was the 33rd significantly hypermethylated CGI in TC [19]. A schematic diagram of the *NPTX2* structure is shown in Figure 1.

Five exons were noted in *NPTX2* mRNA and CGI was detected around exon 1. The array-based methylation status of each CpG site within *NPTX2* is shown in Figure 2. The methylation levels of CGI (from cg09596818 to cg10245879) regions were significantly higher in TC samples than in thymoma samples.

### 2.3. DNA and RNA Extraction

DNA was extracted from frozen tissue and FFPE materials using the QIAamp DNA Mini Kit (Qiagen, Hilden Germany) according to the manufacturer’s instructions. RNA was extracted from frozen tissue using the RNeasy Mini Kit (Qiagen, Hilden, Germany) as outlined in the instruction manual. An ultra-microspectrophotometer (Nanodrop-1000, Thermo Scientific, Wilmington, MA, USA) was used to evaluate the concentration and purity of nucleic acids.

### 2.4. Bisulfite Pyrosequencing

A set of primers designed in the PyroMark Assay Design Software (version 2.0.01.15; Qiagen) was used to amplify bisulfite-treated genomic DNA (Table A2B). We examined the DNA methylation of 4 CpG sites (98246000, 98246002, 98246007, and 98246014) of NPTX2 around cg13314145 (98246002) and cg08315202 (98246007) using pyrosequencing (Figure 1 and Figure 2). PCR product pyrosequencing and methylation quantification were performed using the PyroMark 24 Pyrosequencing System (version 2.0.6; Qiagen) with sequencing primers according to the manufacturer’s instructions. To confirm the relationship between the DNA methylation rates in frozen and FFPE materials, we examined the DNA methylation of *NPTX2* in 9 tumors (7 TC and 2 NECTT) using both frozen and FFPE materials. As shown in Figure A1, a relationship was observed between the *NPTX2* DNA methylation of frozen materials and FFPE materials (Peason’s correlation coefficient test: ρ = 0.692, *p* = 0.039).

### 2.5. Reverse Transcription–Quantitative PCR (RT-qPCR)

RT was conducted with the iScript Reverse Transcription Supermix for RT-PCR (Bio-Rad, Hercules, CA, USA), and qPCR with the KAPA PROBE FAST qPCR Kit (Kapa Biosystems, Wilmington, MA, USA) and *NPTX2* TaqMan Gene Expression Assay (Thermo Fisher Scientific, Inc., Waltham, MA, USA) as described by the manufacturers. Table A2A lists the primers used for qPCR on *NPTX2* and *GAPDH*. Data were normalized to the mRNA level of the internal control *GAPDH*. Human Thymus Total RNA (Takara, Tsu, Japan) was used as the normal thymus control to calculate the mRNA expression level of *NPTX2*.

### 2.6. IHC Staining

The Envision system (ChemMate Envision Kit; Dako, Glostrup, Denmark) was used as described by the manufacturer to stain paraffin-embedded sections for IHC analyses. The primary antibody was a rabbit polyclonal anti-NPTX2 antibody (Sigma-Aldrich, St. Louis, MA, USA; PRS4573) diluted 1:1000 with antibody diluents (Dako, Glostrup Denmark). Dewaxed and dehydrated sections in Dako Real Target Retrieval Solution, pH 9 (Dako, Glostrup, Denmark) were heated using a 2100 retriever (Aptum Biologics, Ltd., Southampton, UK) for antigen activation. Staining scores were calculated as the sum of percentage and intensity scores, with scores > 4 indicating NPTX2 overexpression (Table A3).

IHC scores were independently evaluated by two researchers (KM and KKo).

### 2.7. Statistical Analysis

The distribution of numerical datasets was investigated using the Shapiro–Wilk test. Numerical datasets with a normal distribution were analyzed using a paired *t*-test, the Student’s *t*-test, or Welch’s *t*-test, while those that did not follow a normal distribution were examined by the Wilcoxon signed-rank test or Mann–Whitney test. Continuous data are shown as medians and ranges or interquartile ranges. Multiple comparisons of histology and stage were performed using the Kruskal–Wallis test and Steel–Dwass test. The distribution of age was examined using the unpaired *t*-test, the distributions of sex, histology, and stage by Fisher’s exact test, and the distribution of myasthenia gravis by the chi-squared test. The area under the receiver operating characteristic (ROC) curve (AUC; between 0.5 (chance) and 1.0 (perfect discrimination or accuracy)) reflects the accuracy of the DNA methylation signature used to discriminate between TC and thymomas. A correlation analysis was conducted using Spearman’s rank correlation. Relapse-free survival (RFS) was defined as the time from surgical resection to disease relapse. The Kaplan–Meier method was used to generate survival curves, which were compared with the Log-rank test. Statistical analyses were performed using commercial software programs (GraphPad Prism, version 5.00; GraphPad Software and SPSS, version 24.0; IBM Corp., Armonk, NY, USA). *p*-values < 0.05 indicate a significant difference.

## 3. Results

### 3.1. DNA Methylation Rate of the NPTX2 Gene in TET and Paired Normal Thymic Tissues

The DNA methylation rate of the *NPTX2* gene at four CpG regions and its relationship between thymoma/TC and normal thymic tissues are shown in Figure 3A. The DNA methylation rate did not significantly differ between thymoma and normal thymic tissues (left side, the paired *t*-test, *p* = 0.378), but was significantly higher in TC than in normal thymic tissues (right side, the paired *t*-test, *p* < 0.001).

### 3.2. DNA Methylation Status of the NPTX2 Gene According to the WHO Histological Classification

The median DNA methylation rates of the *NPTX2* gene according to the WHO histological classification of TET were 15.5, 15.6, 20.4, 39.0, 7.3, and 16.0% in A + AB + B1, B2, B3, TC, NECTT, and the normal thymus, respectively (Figure 3B). A significantly higher DNA methylation rate was observed in TC than in the normal thymus, thymoma A + AB + B1, and thymoma B2 (the Kruskal–Wallis test and Steel–Dwass test for multiple comparisons). Similar DNA methylation rates were noted in the thymoma, except for B3, and the normal thymus.

The DNA methylation rate of the *NPTX2* gene in thymoma B3 was midway be- tween those of the normal thymus and TC, while that in NECTT was low.

The accuracy of the DNA methylation signature of *NPTX2* used to discriminate TC from thymomas was assessed using the ROC analysis, with AUC was used as the criterion for accuracy. The ROC curves in Figure 3C showed high sensitivity and specificity for discriminating TC from thymomas (AUC = 0.8007).

### 3.3. DNA Methylation Status of the NPTX2 Gene According to the Masaoka–Koga Stage Classification

The median DNA methylation rates of the *NPTX2* gene in TET classified as stages I, II, III, IVa, and IVb by the Masaoka–Koga staging system were 16.0, 20.8, 22.2, 22.1, and 20.7%, respectively (Figure A2). No significant differences have been observed between stages I +II and III + IV.

### 3.4. mRNA Expression of the NPTX2 Gene in TET and Paired Normal Thymic Tissues

The relationship in terms of the mRNA expression of the *NPTX2* gene between thymoma/TC and the normal thymus is shown in Figure 4A. mRNA expression levels were slightly lower in five thymomas than in the normal thymus (left side, the paired *t*-test, *p* = 0.184), and were also slightly lower in six TC than in the normal thymus (the Wilcoxon signed-rank test, *p* = 0.176).

### 3.5. The mRNA Expression of the NPTX2 Gene According to the WHO Histological Classification

The median mRNA expression levels of *NPTX2* in A + AB + B1, B2, B3, TC, NECTT, and the normal thymus were 0.516, 0.848, 0.89, 0.431, 34.18, and 2.217, respectively (Figure 4B). Although the mRNA expression levels of *NPTX2* were lower in thymomas and TC than in the normal thymus, no significant differences were observed. The mRNA expression level of *NPTX2* in NECTT was very high. ROC curves showing the accuracy of *NPTX2* mRNA expression, used to differentiate TC from thymomas, revealed moderate sensitivity and specificity (AUC = 0.5755, Figure 4C).

### 3.6. mRNA Expression of the NPTX2 Gene According to the Masaoka–Koga Staging System

The median mRNA expression rate of the *NPTX2* gene in TET according to the Masaoka–Koga staging system is shown in Figure A3. The median mRNA expression rates in I, II, III, IVa, and IVb were 1.248, 0.552, 0.607, 0.838, and 2.054, respectively. There was no significant difference in mRNA expression between stages I + II and III + IV.

### 3.7. Relationship between the mRNA Expression and DNA Methylation of NPTX2 in TET

The DNA methylation status and mRNA expression status were investigated in 45 TET (28 thymomas, 14 TC, and 3 NECTT) and 12 normal thymus tissues. The mRNA expression levels inversely correlated with the DNA methylation levels (Spearman’s rank correlation coefficient, ρ = −0.349, *p* = 0.009) (Figure 4D).

### 3.8. Protein Expression of NPTX2 Using IHC

IHC was performed on 39 TET, including 18 thymomas, 18 TC, and 3 NECTT, to assess the NPTX2 protein expression level. Patient characteristics are shown in Table A1. Nerve cells in the brain were used as a positive control in IHC for the NPTX2 protein. The cytoplasmic staining of NPTX2 was observed in nerve cells (Figure 5A(a), but not in thymoma cells (Figure 5A(b)). Tumor cells in NECTT were strongly stained (Figure 5A(d)). Figure 5A(c) shows moderately stained tumor cells in TC. Figure 5B shows the immunoreactivity of NPTX2 in the WHO histological classification. The positive rate was 56% in thymoma, 67% in TC, and 100% in NETCC. The reactivity of NPTX2 was almost constant between TC and thymomas, and was very high and diffuse in all NETCC. Figure 5C shows the relationship between the mRNA expression and immunoreactivity of NPTX2. In total, 18 (58%) out of 31 tumors, in which both mRNA and IHC were measured, showed positive reactivity. Tumors with positive reactivity for NPTX2 expressed slightly higher mRNA levels of NPTX2 (the Mann–Whitney test, *p* = 0.072).

### 3.9. RFS Curve of TET According to DNA Methylation and mRNA Expression Levels

The median follow-up in the present study was 4.52 years (0.15–22.41 years), during which time 7 patients died and 21 had recurrence. RFS analyses of the mRNA expression and DNA methylation levels of NPTX2 were conducted on samples divided into high/low expression and methylation levels using cut-off values from ROC curves. Kaplan–Meier curves of the estimated RFS were generated. The results obtained show that the *NPTX2* DNA methylation levels affected RFS; this value was slightly shorter in patients with high levels than in those with low levels (*p* = 0.102, the Log-rank test, Figure 6A).

NPTX2 mRNA expression did not correlate with RFS (*p* = 0.299, the Log-rank test, Figure 6B).

## 4. Discussion

Genome-wide screening was conducted on aberrantly methylated CGI in patients with TET in the present study, and the results obtained reveal 92 CGI that were significantly hypermethylated in TC relative to thymomas. NPTX2 was identified as the 33rd significantly hypermethylated CGI in TC (19). The focus of the present study was the DNA methylation, mRNA, and protein expression levels of *NPTX2* in TET.

Based on the relevant literature, this appears to be the first study to report the methylation and gene expression levels of *NPTX2,* and their relationships with survival rates in patients with TET.

The DNA methylation of the *NPTX2* gene was significantly higher in TC than in the normal thymus, thymoma A + AB + B1, and thymoma B2. The ROC curves for the accuracy of the *NPTX2* methylation signature differentiate TC from thymomas. Therefore, this has potential as a molecular diagnostic marker for TC. The mRNA expression of *NPTX2* was lower in TC than in the normal thymus. An inverse relationship was observed between mRNA expression levels and methylation levels. The present results indicate that the downregulation of NPTX2 in TC correlated with its aberrant DNA methylation. RFS was shorter in patients with high *NPTX2* DNA methylation levels than in those with low *NPTX2* DNA methylation levels. The aberrant methylation of NPTX2 was associated with a poor prognosis in patients with TC. The DNA methylation rate of the *NPTX2* gene in thymoma B3 was midway between the normal thymus and TC. TC, B3, and thymoma, except B3, were arranged in order of DNA methylation levels. Since B3 thymoma shows more aggressive behavior than other thymomas, the DNA methylation levels were closely linked to the sequence of the malignant potential. However, there were no significant differences in DNA methylation or mRNA expression levels between the early stage (I + II) and advanced stage (III + IV).

On the other hand, NECTT showed very high mRNA and protein expression levels of NPTX2. The present results reveal that TET may be classified into three tumor types—thymoma, TC, and NECTT—according to the DNA methylation and gene expression of *NPTX2*. Thymoma showed similar DNA methylation and gene expression levels to the normal thymus. TC had significantly higher methylation and lower expression levels of *NPTX2* than the normal thymus. NECTT had lower methylation and markedly higher gene expression levels than the normal thymus.

Previous studies have suggested that NPTX2 functions as a tumor suppressor in some tumors, and has an oncogenic function in others. The frequent hypermethylation and significantly lower mRNA and protein expression levels of *NPTX2* were previously reported in pancreatic cancer and glioblastoma than in paired normal tissues [34,35,36]. A correlation was observed between the downregulated expression of *NPTX2* and promoter hypermethylation, which was related to methylation and gene expression levels in TC. Zhang et al. reported that reductions in NPTX2 expression levels in pancreatic cell lines were restored by the addition of 5-aza-2′-deoxycytidine, a DNA methyltransferase inhibitor, and also showed that the ectopic expression of NPTX2 significantly enhanced G0-G1 arrest and cell apoptosis, and suppressed cell proliferation, migration, and invasion [35]. Shukla et al. demonstrated that the treatment of glioblastoma cell lines with 5-aza-2′-deoxycytidine induced the expression of NPTX2 transcripts. Moreover, the exogenous overexpression of NPTX2 inhibited the formation of colonies, promoted apoptosis, suppressed proliferation and anchorage-independent growth, and increased chemosensitivity [36]. These effects of NPTX2 appeared to be attributed to the suppression of the NF-kB pathway through the p53-dependent activation of PTEN, which ultimately inhibited PI3K–AKT–IKKa signaling. Shukla et al. also reported a relationship between the promoter methylation of NPTX2 and a poor prognosis in patients with glioblastoma [36]. The present results show that patients with high *NPTX2* DNA methylation levels had shorter RFS than those with low *NPTX2* DNA methylation levels. Therefore, NPTX2 may act as a tumor suppressor in not only pancreatic cancer and glioblastoma, but also TC.

NPTX2 mRNA and protein expression levels were previously reported to be significantly higher in colorectal carcinoma (CRC), clear cell renal cell carcinoma (ccRCC), and neuroblastoma than in normal tissue [31,32,37]. Xu et al. demonstrated an increase in NPTX2 expression levels with the progression of CRC, which was associated with a poor prognosis. The findings of in vitro experiments have shown that CRC growth and liver metastasis were both promoted by NPTX2 through the activation of the canonical Wnt/β-catenin pathway via FZD6 [37]. Other studies revealed correlations between high NPTX2 mRNA and protein expression levels and the proliferation and metastasis of ccRCC and neuroblastoma [31,32]. Roemeling et al. detected the overexpression of NPTX2 in ccRCC primary tumors and metastases, and showed that it promoted tumor cell viability and migration by interacting with the AMPA receptor subunit GluR4 [32]. Bartolini et al. showed the upregulated expression of NPTX2 in neuroblastoma, and found a correlation between high NPTX2 levels and poor overall survival [31]. The Human Protein Atlas revealed moderate to strong cytoplasmic positivity for NPTX2 in rare carcinoid and renal cancers, while all remaining cancer tissues were negative [38]. Collectively, these findings suggest that NPTX2 has as oncogenic function in CRC, ccRCC, and neuroblastoma, as well as in NECTT.

The present study had some limitations that need to be addressed. (1) Since TC and NETCC are very rare tumors, we used not only frozen materials, but also FFPE materials in the DNA methylation analysis. Although we examined DNA methylation in 25 TC samples, we only analyzed six NETCC samples. To confirm that “NPTX2 may have an oncogenic function in NECTT”, we need more NETCC samples. (2) Since there are no thymoma or NETCC cell lines and only a few TC cell lines, we were unable to perform in vitro experiments, including re-expression by a methylation inhibitor treatment (5-aza-2′-deoxycytidine), or assays on colony formation, apoptosis, and proliferation using the exogenous up- and downregulation of NPTX2. (3) Thymomas are classified by epithelial cell morphology and the lymphocyte-to-epithelial cell ratio. Since the lymphocyte-to-tumor cell ratio in AB, B1, and B2 thymomas is high, the presence of lymphocytes may have affected the promoter methylation rate.

## 5. Conclusions

The DNA methylation rate of the *NPTX2* gene was significantly higher in TC than in the normal thymus, thymoma A + AB + B1, and thymoma B2. The mRNA expression level of *NPTX2* was lower in TC than in the normal thymus. An inverse relationship was observed between mRNA expression levels and methylation levels. RFS was shorter in patients with high *NPTX2* DNA methylation levels than in those with low *NPTX2* DNA methylation levels. On the other hand, NECTT had lower methylation and markedly higher gene expression levels than the normal thymus. These results suggest that NPTX2 functions as a tumor suppressor in TC and has an oncogenic function in NECTT.

## Figures and Tables

**Figure 1 cancers-16-00329-f001:**
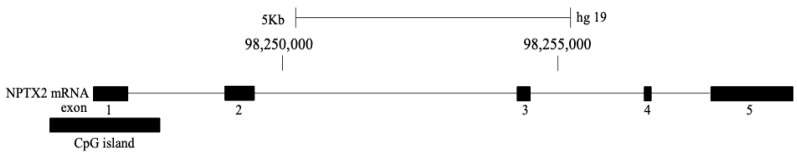
Schematic structure of the human *NPTX2* structure. The mRNA of *NPTX2* has 5 exons. The CGI is located around exon 1. “UCSC Genome Browser (Human GRCh 37/hg19)”. https://genome.ucsc.edu/cgi-bin/hgGateway?hgsid=599344653_C3wSlv7f4i4BPmj1dN500QJ2IlmW (accessed on 30 August 2018).

**Figure 2 cancers-16-00329-f002:**
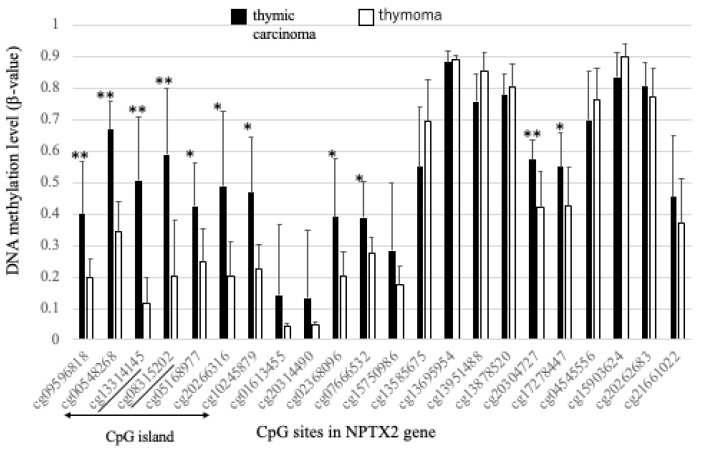
DNA methylation levels. The vertical axis shows the average β-value for the DNA methylation level of each CpG site between TC (black bar) and B3 thymomas (white bar). Mean ± standard deviation (SD). The target regions for the pyrosequencing analysis are located between cg13314145 and cg08315202 (underlined). The asterisk shows CpG sites with a significantly different (** *p* < 0.01, * *p* < 0.05) methylation rate between TC and thymoma.

**Figure 3 cancers-16-00329-f003:**
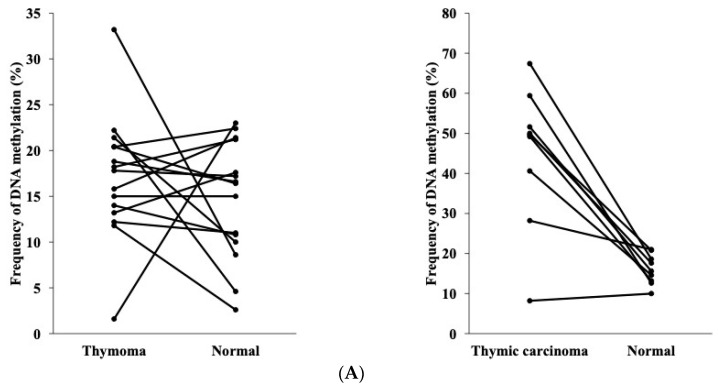
(**A**) DNA methylation rate of the *NPTX2* gene in thymomas (n = 15) and paired normal thymic tissues (n = 15) (**Left**), and in TC (n = 9) and paired normal thymic tissues (n = 9) (**Right**) (the paired *t*-test). Samples from the same patient are linked with straight lines. (**B**) DNA methylation rate of the *NPTX2* gene in TET according to the World Health Organization histological classification. The upper and lower ends of the whiskers show the upper and lower extremes, the upper and lower edges of the boxes show the upper (75th) and lower (25th) quartiles, and horizontal lines across each box show medians. Data were analyzed using the Kruskal–Wallis test and Steel–Dwass test for multiple comparisons. Asterisks indicate significant differences between groups by steel-Dwass. (** *p* < 0.01) (**C**) Receiver operating characteristic curve (ROC) was used for determining the accuracy of the methylation signature used for TC detection from all tumors using *NPTX2* methylation levels.

**Figure 4 cancers-16-00329-f004:**
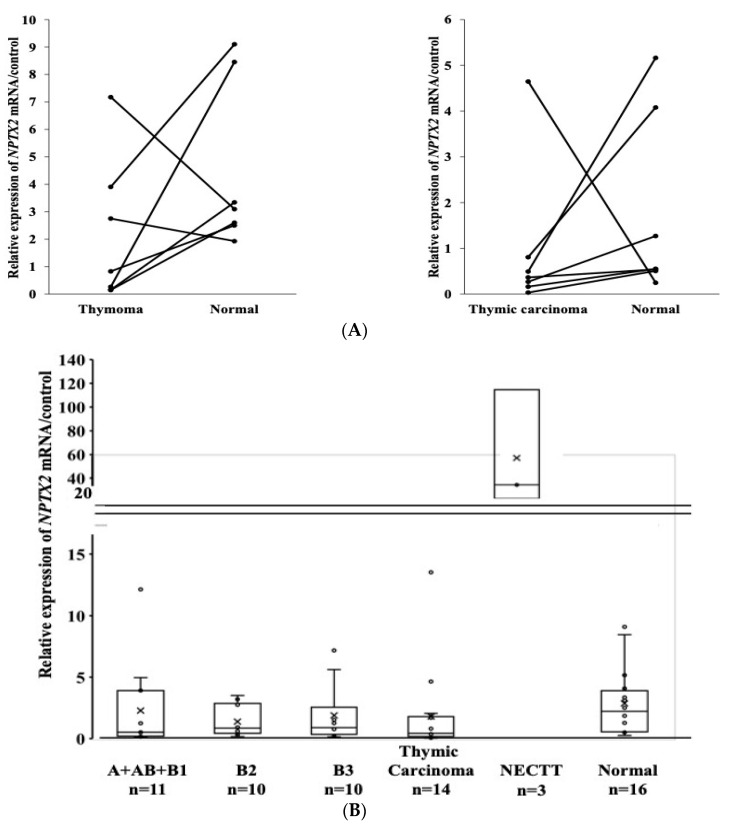
(**A**) The mRNA levels of the *NPTX2* gene in thymomas and paired normal thymic tissues (n = 7, (**Left**)), and in TC and paired normal thymic tissues (n = 7, (**Right**)) (the Wilcoxon signed-rank test). Samples from the same patient are linked with straight lines. (**B**) *NPTX2* mRNA expression in TET according to the WHO histological classification (A + AB + B1, B2, B3, and TC). The upper and lower ends of the whiskers show the upper and lower extremes, the upper and lower edges of the boxes show the upper (75th) and lower (25th) quartiles, and horizontal lines across each box show medians. Data were analyzed using the Kruskal–Wallis test and Steel–Dwass test. (**C**) Receiver operating characteristic curve (ROC) used to determine the accuracy of the methylation signature used for TC detection from all tumors in *NPTX2* mRNA expression. (**D**) Correlation between the DNA methylation and mRNA expression of *NPTX2* in 33 tumors and 12 normal thymus tissues. Data were analyzed via Spearman’s rank correlation.

**Figure 5 cancers-16-00329-f005:**
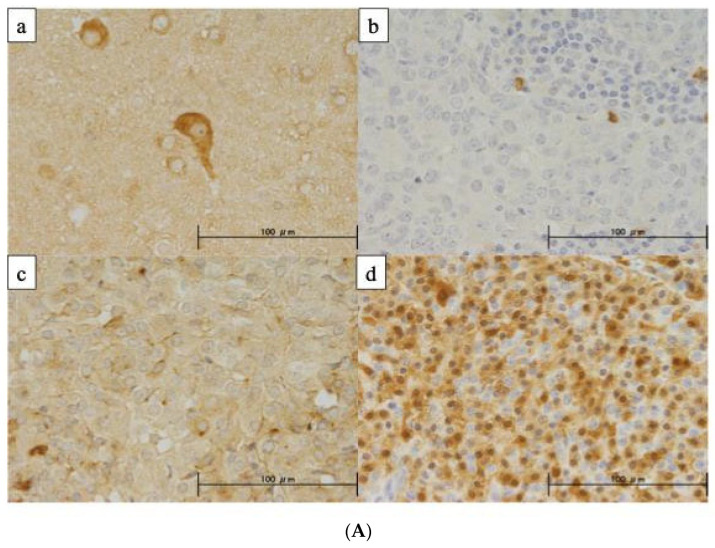
(**A**) Representative images of the immunohistochemically detected NPTX2 protein in tumors. Scale bar, 100 µm. (**a**) Nerve cells in the brain were used as a positive control in IHC for the NPTX2 protein. The cytoplasmic staining of NPTX2 was observed in nerve cells. (**b**) No staining was noted in thymoma cells. (**c**) Tumor cells in TC were moderately stained. (**d**) Tumor cells in NECTT were strongly and diffusely stained. (**B**) Positive rate of NPTX2 protein expression in TET according to the WHO histological classification. The positive rate was 56% in thymoma, 67% in TC, and 100% in NETCC. Thymoma type A, B1, B2, and B3, TC; thymic carcinoma, NECTT; neuroendocrine tumors of the thymus (**C**) Relationship between the mRNA expression and immunoreactivity of NPTX2. We performed 39 TET for IHC. Both mRNA and IHC were measured in 31 tumors. In total, 18 (58%) out of 31 tumors showed positive reactivity.

**Figure 6 cancers-16-00329-f006:**
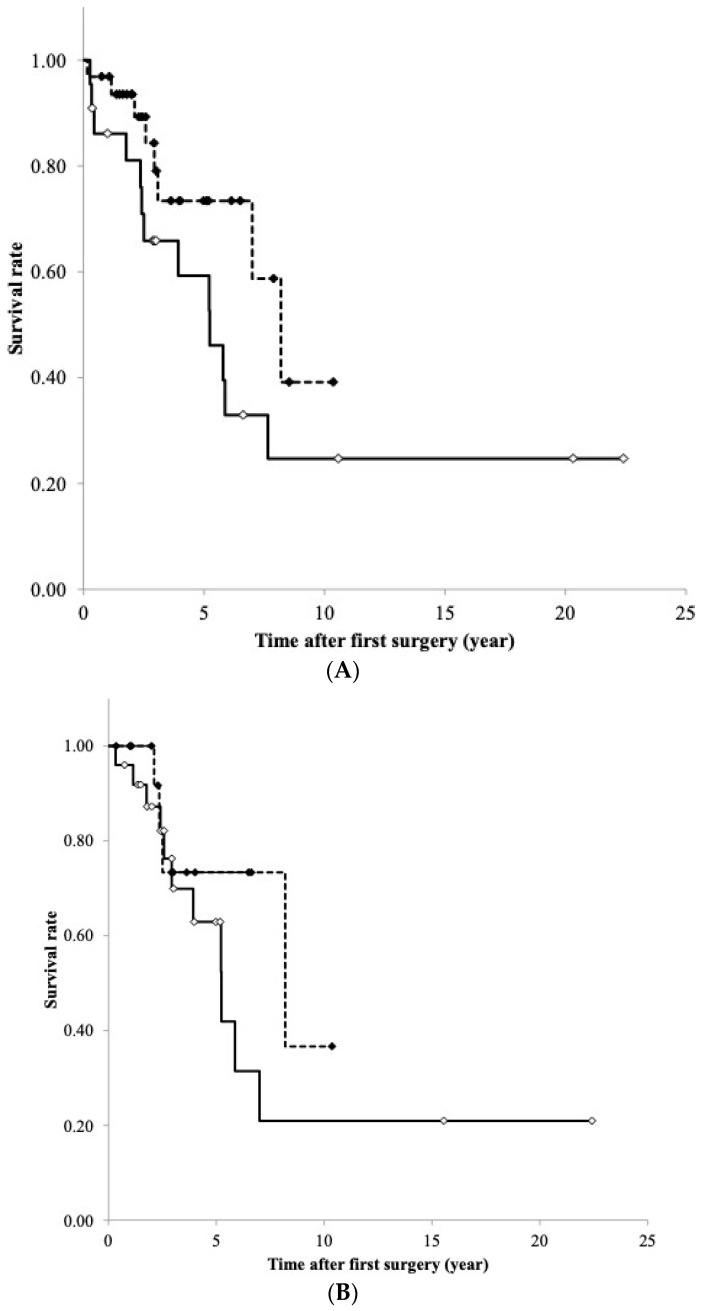
(**A**) Relapse-free survival curves of TET patients with higher and lower DNA methylation levels of NPTX2. The median value (23.8) for the frequency of the DNA methylation of NPTX2 was used to divide patients into the hypermethylated and hypomethylated groups. The dotted line shows the lower DNA methylation of NPTX2 (n = 32), while the solid line shows the higher DNA methylation of NPTX2 (n = 22). *p*-values were calculated using the Log-rank test. (**B**) Relapse-free survival curves of patients with TET with higher and lower levels of NPTX2 mRNA expression. The median value (0.479) of NPTX2 mRNA expression was used to divide patients into the high- and low-expression groups. The dotted line shows the lower expression of NPTX2 (n = 16), and the solid line shows the higher expression of NPTX2 (n = 25). *p*-values were calculated using the Log-rank test.

**Table 1 cancers-16-00329-t001:** Clinical and pathological characteristics of patients.

		Total	Pyrosequencing Analysis	RT-PCR Analysis	IHC Analysis
Gender					
	male	31 (46.3%)	29 (45.3%)	23 (47.9%)	17 (43.6%)
	female	36 (53.7%)	35 (54.7%)	25 (52.1%)	22 (56.4%)
Age (year)		61.2 ± 11.9 #	61.1 ± 12.1	60.6 ± 12.2	60.7 ± 11.5
		(28~84) !	(28~84)	(28~84)	(28~80)
WHO histological classification					
Thymoma		36 (53.7%)	33 (51.6%)	31 (64.6%)	18 (46.2%)
	A	6 (9.0%)	5 (7.8%)	5 (10.4%)	3 (7.7%)
	AB	3 (4.5%)	2 (3.1%)	1 (2.1%)	0 (0.0%)
	B1	5 (7.5%)	5 (7.8%)	5 (10.4%)	2 (5.1%)
	B2	10 (14.9%)	10 (15.6%)	10 (20.8%)	7 (17.9%)
	B3	12 (17.9%)	11 (17.2%)	10 (20.8%)	6 (15.4%)
Thymic carcinoma *		25 (37.3%)	25 (39.1%)	14 (29.2%)	18 (46.2%)
NECTT		6 (9.0%)	6 (9.4%)	3 (6.3%)	3 (7.7%)
	typical carcinoid	4 (6.0%)	4 (6.3%)	2 (4.2%)	2 (5.1%)
	atypical carcinoid	2 (3.0%)	2 (3.1%)	1 (2.1%)	1 (2.6%)
Masaoka–Koga staging system					
	I	16 (23.9%)	14 (21.9%)	13 (27.1%)	9 (23.1%)
	II	22 (32.8%)	21 (32.8%)	18 (37.5%)	15 (38.5%)
	III	13 (19.4%)	13 (20.3%)	6 (12.5%)	6 (15.4%)
	IVA	8 (11.9%)	8 (12.5%)	6 (12.5%)	4 (10.3%)
	IVB	8 (11.9%)	8 (12.5%)	5 (10.4%)	5 (12.8%)
Myasthenia gravis					
	-	55 (82.1%)	53 (82.8%)	37 (77.1%)	32 (82.1%)
	+	12 (17.9%)	11 (17.2%)	11 (22.9%)	7 (17.9%)

* Two case of combined small-cell carcinoma, one case of combined leiomyosarcoma. # mean ± standard deviation (SD). ! (minimum~maximum). RT, reverse transcription. IHC, immunohistochemistry. NECTT, neuroendocrine tumor of the thymus.

## Data Availability

The datasets used and/or analyzed during the current study are available from the corresponding author on reasonable request.

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
