# Peer review of "Aberrant DNA Methylation of NPTX2 as an Indicator of Malignant Behavior in Thymic Epithelial Tumorsâ€"

_cancers, 2024, doi:10.3390/cancers16020329_

Round 1
Reviewer 1 Report
Comments and Suggestions for Authors
Takizawa and colleagues analysed the DNA methylation profile of NPTX2 in Thymic Epithelial Tumours (TETs) and found that NTPX2 is hypermethylated in Thymic carcinomas (TC) in comparison to normal thymus and thymomas (except for B3). Additionally, they found that NPTX2 expression is lower in TC than normal thymus and that hypermethylation of NPTX2 is associated with decreased relapse free survival. In contrast with TC, NPTX2 is hypomethylated and over expressed in neuroendocrine tumours of the thymus (NECTT).
These findings are interesting but the manuscript suffers from the very poor presentation of the data. Specifically:
- There is no flow in the figures that are assembled in random order. Figure 1A and 1B is followed by figure A1. Figure 2 is followed by figure A2, figure 3 is followed by figure A3 etc. Moreover, there is no matching between figures and figure legends with legend 3C describing fig 3D, legend 3B describing 3C etc.
- Most of the figure legends also do not provide sufficient information about the figures.
- The other major problem of the manuscript is that it is not clear how many specimens were analysed to generate the data. The authors present the limitation that they did not have frozen sections from all the tumors and this is understandable. However, the numbers presented in the figures are rather confusing. For example, in figure 2A we have 15 thymomas and 9 TC whereas in figure 2B we have methylation data from all the tumors. Similarly, in figure 3A we have data from 7 thymomas and 7 TC whereas in figure 3B data are generated from different number of specimens. How are these discrepancies explained?
- Figure 4: a) How does the staining score method listed in Table A3 and methods section lead to the data depicted figure 4e? Please explain figure 4f and why we have data from 31 patients when IHC was performed on 39 samples.
- Figure 1: a) Which software was used to map NPTX2? b) Error bars in figure 1b?
- Figure 5: p-values?
Comments on the Quality of English Language
It is difficult to read the manuscript. The extensive use of hyphens in individual words does not help at all.
Author Response
I thank the reviewers for their thoughtful suggestions and insights.
The manuscript has been revised, and necessary changes have been made in accordance with the reviewers’ suggestions.
Our point-by-point responses to the reviewers’ comments are given below.
Takizawa and colleagues analysed the DNA methylation profile of NPTX2 in Thymic Epithelial Tumours (TETs) and found that NTPX2 is hypermethylated in Thymic carcinomas (TC) in comparison to normal thymus and thymomas (except for B3). Additionally, they found that NPTX2 expression is lower in TC than normal thymus and that hypermethylation of NPTX2 is associated with decreased relapse free survival. In contrast with TC, NPTX2 is hypomethylated and over expressed in neuroendocrine tumours of the thymus (NECTT).
These findings are interesting but the manuscript suffers from the very poor presentation of the data. Specifically:
- There is no flow in the figures that are assembled in random order. Figure 1A and 1B is followed by figure A1. Figure 2 is followed by figure A2, figure 3 is followed by figure A3 etc. Moreover, there is no matching between figures and figure legends with legend 3C describing fig 3D, legend 3B describing 3C etc.
→Figure A1, A2 and A3 are “supplement figures”.
These Figures move to the end of manuscript.
There is flow in the figures. Figure 1A, 1B, →2A, 2B, 2C→3A, 3B, 3C, 3D→4A, 4B, 4C→5A, 5B
We want to change the figure number to be better flow.
Figure 4 is changed to Figure 4A.
Figure 4-a→Figure 4A-a, Figure 4-b→Figure 4A-b, Figure 4-c→Figure 4A-c, Figure 4-d→Figure 4A-d,
Figure 4-e are changed to Figure 4B.
Figure 4-f are changed to Figure 4C.
- Most of the figure legends also do not provide sufficient information about the figures.
We revised Figure legends.
→We added “UCSC Genome Browser (Human GRCh 37/hg19)”
https://genome.ucsc.edu/cgi-bin/hgGateway?hgsid=599344653_C3wSlv7f4i4BPmj1dN500QJ2IlmW in Figure 1A legends. (green sentence).
We added “The asterisk shows CpG sites with a significantly different methylation rate between TC and thymoma (**P <0.01, *P <0.05).” in Figure 1B legends (green sentence).
We added “Samples from the same patient are linked with straight lines.” in Figure 2A and 3A legends. (green sentence)
We added “Receiver operating characteristic curve (ROC) for the accuracy of the methylation signature for TC detection from all tumors in NPTX2 methylation levels.“ in Figure 2C and “Receiver operating characteristic curve (ROC) for the accuracy of the methylation signature for TC detection from all tumors in NPTX2 mRNA expression” in Figure.3C. (green sentence)
Figure 4A legends are “Representative images of the immunohistochemically detected NPTX2 protein in tumors. Scale bar, 100 µm. (a)Nerve cells in the brain were used as a positive control in IHC for the NPTX2 protein. The cytoplasmic staining of NPTX2 was observed in nerve cells. (b) No staining was noted in thymoma cells. (c) Tumor cells in TC were moderately stained. (d) Tumor cells in NECTT were strongly and diffusely stained.” (green sentence)
We added “The positive rate was 56% in thymoma, 67% in TC, and 100% in NETCC.” In Figure 4B legends. (green sentence)
We added “We performed 39 TET for IHC. Both of mRNA expression measure and IHC staining cases is 31. Eighteen (58%) out of 31 tumors showed positive reactivity.” In Figure 4C legends. (green sentence)
- The other major problem of the manuscript is that it is not clear how many specimens were analysed to generate the data. The authors present the limitation that they did not have frozen sections from all the tumors and this is understandable. However, the numbers presented in the figures are rather confusing. For example, in figure 2A we have 15 thymomas and 9 TC whereas in figure 2B we have methylation data from all the tumors. Similarly, in figure 3A we have data from 7 thymomas and 7 TC whereas in figure 3B data are generated from different number of specimens. How are these discrepancies explained?
→We had 15 frozen TC tissues and 3 frozen NECTT. We thought that DNA methylation rate and mRNA levels of TC and NECTT is very important in this study. We added paraffin-embedded sample of 10 TC and 3 NECTT. Figure A1 (supplementary figure) showed there are correlation between DNA methylation levels of the NPTX2 in both frozen and FFPE materials of nine tumors (Pearson’s correlation coefficient test: ρ=0.692, P=0.039).
→In DNA methylation assay (figure 2A), the number of the tumor tissues and paired normal thymic tissues is 24. the number of tumors without paired normal thymic tissues is 40. In figure 2A, we used 24 tumor tissues (15 thymomas and 9 TCs) and 24 paired normal thymic tissues. In figure 2B, we used 40 tumors without paired normal thymic tissues and 24 tumor tissues and paired normal thymic tissues. (Tumor=64 cases, normal=24 cases)
→In mRNA assay (figure 3A), we used 14 tumor tissues (7 thymomas and 7 TCs) and 14 paired normal thymic tissues. In figure 3B, we used 34 tumors without paired normal thymic tissues and 14 tumor tissues and paired normal thymic tissues and 2 normal thymic tissue without tumor. (Tumor=34+14=48, normal=14+2=16)
We showed it in Table A1 (supplementary Table, the patients lists) in detail.
- Figure 4: a) How does the staining score method listed in Table A3 and methods section lead to the data depicted figure 4e? Please explain figure 4f and why we have data from 31 patients when IHC was performed on 39 samples.
→2.6 In IHC staining in Method section (Page 6, lines181-182) We described “Staining scores were calculated as the sum of percentage and intensity scores, with scores >4 indicating NPTX2 overexpression (Table A3)”.
→We performed 39 TET for IHC. Both of mRNA expression measure and IHC staining cases is 31. Although 8 cases were performed for IHC, they did not measure mRNA expression.
- Figure 1: a) Which software was used to map NPTX2? b) Error bars in figure 1b?
→We made genomic map using “UCSC Genome Browser (Human GRCh 37/hg19)”
https://genome.ucsc.edu/cgi-bin/hgGateway?hgsid=599344653_C3wSlv7f4i4BPmj1dN500QJ2IlmW
We added standard deviation in each CpG site of Figure 1B.
- Figure 5: p-values?
→We described “RFS; it was slightly shorter in patients with high levels than in those with low levels (P=0.102, the Log-rank test, Fig. 5A). NPTX2 mRNA expression did not correlate with RFS (P=0.299, the Log-rank test, Fig. 5B).” (Page 14, Lines331-334).
Comments on the Quality of English Language
It is difficult to read the manuscript. The extensive use of hyphens in individual words does not help at all.
→I am sorry that there are many hyphens in words. I revised them (the revised portions are green words).
Reviewer 2 Report
Comments and Suggestions for Authors
The main strength of the manuscript is the rarity of samples the authors managed to obtain for the analysis. Although I fully understand the difficulties of analyzing a small number of samples, I find that the manuscript could be sufficiently improved prior to publication without further sample collection.
Major concerns:
1. Table 1 is virtually unreadable. I presume technical problems were involved. Also, it would be beneficial if statistical significance (by means of χ2 for IHC and t-test for rt-qPCR) could be included for comparing differences between groups
2. It is not clear what the percentages in RT-qPCR analysis stand for. The rt-qPCR values should be presented as scale variables, either as mean expression levels±SD if distribution is normal; or median levels±25th percentile.
3. Numerous typos are present throughout the manuscript
4. Why was pentraxin-2 analysed when it is, as the authors claim, the 33rd significantly hypermethylated CGI in TC? Why was not another more significant gene analysed?
5. There seem to be some serious mistakes regarding the numbering of figures. Figures 1.A and 1.B are ok, but what is figure A1 is not. Please revise.
6. Although there is a significance in statistics, the number of samples in FigA1 (which is obviously wrongly enumerated) is very low, while the regression line is not convincing. Can the authors explain the low methylation levels in two FFPE samples compared to frozen section samples?
7. In Fig4 there is no figure legend. It also seems that there are some major outliers in the positive group. The authors should reanalyze these samples and redo the analysis.
8. Given its function described in the Introduction section, how do the authors explain the role of hypermethylation of NPTX2 in thymic carcinoma? This should be more elaborated in the Discussion section
Author Response
I thank the reviewers for their thoughtful suggestions and insights.
The manuscript has been revised, and necessary changes have been made in accordance with the reviewers’ suggestions.
Our point-by-point responses to the reviewers’ comments are given below.
The main strength of the manuscript is the rarity of samples the authors managed to obtain for the analysis. Although I fully understand the difficulties of analyzing a small number of samples, I find that the manuscript could be sufficiently improved prior to publication without further sample collection.
Major concerns:
- Table 1 is virtually unreadable. I presume technical problems were involved. Also, it would be beneficial if statistical significance (by means of χ2 for IHC and t-test for rt-qPCR) could be included for comparing differences between groups
→I am very sorry that Table 1 was poor findings. I revised it.
As most of the patients in each group (total case, pyrosequencing case, RT-PCR case and IHC case) were piled up, we cannot perform a statistical analysis such as, t-test, χ2 etc. except descriptive statistics.
- It is not clear what the percentages in RT-qPCR analysis stand for. The rt-qPCR values should be presented as scale variables, either as mean expression levels±SD if distribution is normal; or median levels±25th percentile.
→The percentages in pyrosequencing, RT-PCR analysis and IHC analysis in Table 1 present the percentages of case number in each analysis. For example, in RT-PCR analysis male is 23 cases (47.9%) and female in 25 cases (52.1%). In age 60.6±12.2 presents mean±standard deviation (SD).
- Numerous typos are present throughout the manuscript
→I am sorry that there are many hyphens in words. I revised them (the revised portions are green words).
- Why was pentraxin-2 analysed when it is, as the authors claim, the 33rd significantly hypermethylated CGI in TC? Why was not another more significant gene analysed?
→We identified 92 CGI using comprehensive DNA methylation analysis. We looked for cancer-related genes in these genes using references. Several genes- GHSR(16th), GNG4(7th), HOXD9(23rd), SALL3(26th), GAD1(4th), NPTX2(33rd) and MT1(39th) were selected. These genes have been examined and made manuscripts.
- Kishibuchi, R.; Kondo, K.; Soejima, S.; Tsuboi, M.; Kajiura, K.; Kawakami, Y.; Kawakita, N.; Sawada, T.; Toba, H.; Yoshida, M.; et al. DNA methylation of GHSR, GNG4, HOXD9 and SALL3 is a common epigenetic alteration in thymic carcinoma. Int J Oncol 2020, 56, 315-326.
- Soejima, ; Kondo, K.; Tsuboi, M.; Muguruma, K.; Tegshee, B.; Kawakami, Y.; Kajiura, K.; Ka- wakita, N.; Toba, H.; Yoshida, M.;et al.: GAD1 expression and its methylation as indicators of malignant behavior in thymic epithelial tumors, Oncology Letters, 2021, 21, 483,.
- 2019 World Conference on Lung Cancer Sep 7-10 2019 in Barcelona, Spain, and published as abstract (no. MA20.03) in Journal of Thoracic Oncology Volume 14, ISSUE 10, S331: 2019.
- There seem to be some serious mistakes regarding the numbering of figures. Figures 1.A and 1.B are ok, but what is figure A1 is not. Please revise.
→Figure A1, A2 and A3 shows “supplement figures”.
These Figures move to the end of manuscript.
There is flow in the figures.
Figure 1A, 1B, →2A, 2B, 2C→3A, 3B, 3C, 3D→4A, 4B, 4C→5A, 5B
We want to change the figure number to be better flow.
Figure 4 is changed to Figure 4A.
Figure 4-a→Figure 4A-a, Figure 4-b→Figure 4A-b, Figure 4-c→Figure 4A-c, Figure 4-d→Figure 4A-d,
Figure 4-e are changed to Figure 4B.
Figure 4-f are changed to Figure 4C.
- Although there is a significance in statistics, the number of samples in FigA1 (which is obviously wrongly enumerated) is very low, while the regression line is not convincing. Can the authors explain the low methylation levels in two FFPE samples compared to frozen section samples?
We performed Pearson’s correlation coefficient test again. The result was same: ρ=0.692, P=0.039. We deleted the regression line. As the Reviewer pointed out, two FFPE samples showed lower DNA methylation rate comparing to frozen samples. We suspect that the DNA quality of FFPE samples was worse. However, as TC and NECTT are very rare tumor, it is necessary to use FFPE samples.
- In Fig4 there is no figure legend. It also seems that there are some major outliers in the positive group. The authors should reanalyze these samples and redo the analysis.
We want to change the figure number to be better flow.
Figure 4 is changed to Figure 4A.
Figure 4-a→Figure 4A-a, Figure 4-b→Figure 4A-b, Figure 4-c→Figure 4A-c, Figure 4-d→Figure 4A-d,
Figure 4-e are changed to Figure 4B.
Figure 4-f are changed to Figure 4C.
Figure legend of Figure 4 is:
Figure 4. A) Representative images of the immunohistochemically detected NPTX2 protein in tumors. Scale bar, 100 µm. (a)Nerve cells in the brain were used as a positive control in IHC for the NPTX2 protein. The cytoplasmic staining of NPTX2 was observed in nerve cells. (b) No staining was noted in thymoma cells. (c) Tumor cells in TC were moderately stained. (d) Tumor cells in NECTT were strongly and diffusely stained.
As the Reviewer pointed out, there are some outliers in Figure 4C. We performed statistical analysis (the Mann-Whitney test) again. The result was same (P=0.072). When we excluded highest outlier, the result was P=0.107.
- Given its function described in the Introduction section, how do the authors explain the role of hypermethylation of NPTX2 in thymic carcinoma? This should be more elaborated in the Discussion section
→We described that the normal function of NPTX2 is a protective mechanism against excitotoxicity in neural system (Page 2, Lines 83-89). However, I do not know whether this function involve a carcinogenesis and cancer progression or not.
Pancreatic cancer and glioblastoma shows high DNA methylation and lower mRNA and protein expression and a poor prognosis. It is similar to TC. We think that the similar mechanism for carcinogenesis and progression plays as tumor suppressor gene in TC. In pancreatic cancer and glioblastoma, several in vitro researches about tumor suppressor function are performed using cell lines. Zhang et al. suggested that NPTX2 may be related to apoptotic molecules such as, bax, bcl-2, etc (Page 16, Lines 390-394). Shukla et al. speculated that the DNA methylation of NPTX2 is related to NF-kB activity by inhibiting AKT through a p53-PTEN dependent pathway (Page 17 Lines 394-402). As the number of cell lines of TC is very small for in vitro experiments, it may be difficult to clarify the role of hypermethylation in TC.
Round 2
Reviewer 2 Report
Comments and Suggestions for Authors
I find the manuscript is now sufficiently improved to be accepted for publication.
Comments on the Quality of English LanguageEnglish language is fine
Author Response
Comments and Suggestions for Authors
I find the manuscript is now sufficiently improved to be accepted for publication.
Comments on the Quality of English Language
English language is fine.
→Thank you very much for evaluating our research and manuscript.